# Community structure of actively growing bacteria in a coastal fish-farming area

**Akito Taniguchi**[1]*, **Mitsuru Eguchi**[1,2]

**1** Department of Fisheries, Faculty of Agriculture, Kindai University, Nara, Japan, **2** Agricultural Technology and Innovation Research Institute, Kindai University, Nara, Japan

* htrakito@nara.kindai.ac.jp

**Data Availability Statement:** All nucleotide sequences are available from the DNA Data Bank of Japan nucleotide sequence database (https://www.ddbj.nig.ac.jp/index-e.html) (accession numbers LC314448 to LC314476). The other relevant data

## Abstract

In fish-farming areas, copious amounts of organic matter are released into the surrounding environment. Although it is well-known that bacterial community structures and activities are tightly coupled with organic conditions in the environment, actively growing bacteria (AGB) species that are responsible are still largely unknown. Here, we determined seasonal variations in the community structures of free-living and particle-attached AGB in surface and bottom seawater, and also in the easily resuspendable sediment boundary layer. Accordingly, we used bromodeoxyuridine (BrdU) magnetic bead immunocapture and PCR-denaturing gradient gel electrophoresis (BUMP-DGGE) analysis. Whereas overall bacterial communities in the resuspendable sediment were quite different from those of the free-living and particle-attached bacteria, the AGB community structures were similar among them. This result suggests that sediment resuspension in aquaculture environments functions as an organic source for bacteria in the water column, and that bacterial species contributing to the environmental capacity and carbon cycle are limited. We identified 25 AGB phylotypes, belonging to Alphaproteobacteria (*Roseobacter* clade, nine phylotypes), Gammaproteobacteria (five phylotypes), Deltaproteobacteria (one phylotype), Bacteroidetes (seven phylotypes), and Actinobacteria (three phylotypes). Among them, some AGB phylotypes appeared throughout the year with high frequency and were also identified in other coastal environments. This result suggests that these species are responsible for the environmental capacity and carbon cycle, and are key species in this fish-farming area, as well as other coastal environments.

## Introduction

In aquaculture environments, copious amounts of organic matter, such as feed wastage and fish feces are released into the surrounding environment [1]. In addition to allochthonous organic matter, the presence of autochthonous organic matter, from phytoplankton photosynthesis, is also considerable [2]. Sometimes, this organic load has led to various problems, such as fish disease, eutrophication, and development of dysoxic/anoxic conditions in the bottom sediment, followed by toxic gas production [3]. For sustainable aquaculture, activity must be maintained within the environmental capacity, which is defined as the ability of the

are within the manuscript and its Supporting Information files.

**Funding:** This study was supported by the Global COE program "International education and research center for aquaculture science of bluefin tuna and other cultured fish" from the Ministry of Education, Culture, Sports, Science and Technology of Japan. The funders had no role in study design, data collection and analysis, decision to publish, or preparation of the manuscript.

**Competing interests:** The authors have declared that no competing interests exist.

environment to accommodate a particular activity or rate of activity without unacceptable impact [4]. Among the various factors that define the environmental capacity, bacterial processes, such as organic matter degradation and conversion, are critical.

Particles are more abundant in aquaculture environments where organic loading is higher than in other marine environments [2, 5]. Therefore, the contribution of particle-attached bacteria becomes more important. Particles are hotspots for bacterial abundance, activity, and diversity [6], and particle-attached bacteria should thus significantly contribute to the carbon cycle in marine environments [7]. Yoshikawa et al. [8, 9] clearly showed that hydrolysis and mineralization are higher in sinking particles than in seawater and bottom sediments, suggesting that sinking particles are one of the key sites of microbial hydrolysis and mineralization in aquaculture environments. Previous studies analyzing the community structures of particle-attached bacteria [10–13] showed that Gammaproteobacteria and Bacteroidetes frequently dominated these particles. Further, most studies have focused on describing the general phylogenetic affiliations of these particle-attached bacteria, but information on actively growing species is limited.

Bacteria are tightly coupled with organic conditions in the environment and change rapidly in terms of activity and community structure [14, 15]. It is thought that actively growing bacteria (AGB) contribute to the degradation of organic carbon because they require increased levels of organic carbon to maintain their cell production [16]. Simultaneously, AGB are easily lyzed by viruses and/or grazed on by heterotrophic or mixotrophic protists [17, 18]. Previously, the bromodeoxyuridine (BrdU) technique has been used to identify AGB. BrdU, which is a halogenated nucleoside and thymidine analog, is used to monitor bacteria that are actively synthesizing DNA. Thus, bacteria that incorporate BrdU into their DNA can be considered AGB. BrdU-incorporated DNA can then be detected with specific antibodies [19–21]. An immunocapture technique with magnetic-bead-conjugated antibodies has been utilized to determine the phylogenetic affiliations of active bacterial groups in soil and aquatic environments [22, 23]. For example, the diversity and spatio-temporal variability of AGB were investigated in coastal and oceanic environments using BrdU magnetic bead immunocapture and PCR-denaturing gradient gel electrophoresis (BUMP-DGGE) [16, 24, 25]. BrdU is thus one of the most powerful tools to investigate active bacterial diversity [26] that is responsible for the carbon cycle in aquaculture environments. Using this technique, it is possible to identify key bacterial species that directly contribute to degradation and/or mineralization of organic matter and to ensure sustainable aquaculture environments.

The objectives of the present study were to utilize BrdU to determine the phylogenetic affiliations of AGB in the subsurface (ca. 10 m) of a coastal shallow fish-farming area. We specifically analyzed the free-living and particle-attached AGB in the water column. Yoshikawa et al. [9] suggested that microbial degradation of organic matter is stimulated by the resuspension of bottom sediments because of the high hydrolysis rate of resuspendable sediments. Therefore, we also investigated AGB in the resuspendable sediment, which represents an easily resuspended boundary layer, with gentle shaking. This is the first study that the AGB, not inactive and/or dead bacterial cells, which can directly contribute to the organic matter cycle of fish-farming areas, have been identified.

## Materials and methods

### Seawater and resuspendable sediment sampling

Bimonthly sampling was performed from May 2009 to March 2010 in fish cages (red seabream *Pagrus major* was mainly cultured; water depth was approximately 10 m) belonging to Kindai University in Tanabe Bay, Japan (Fig 1). In this area, fish have been cultured for more than

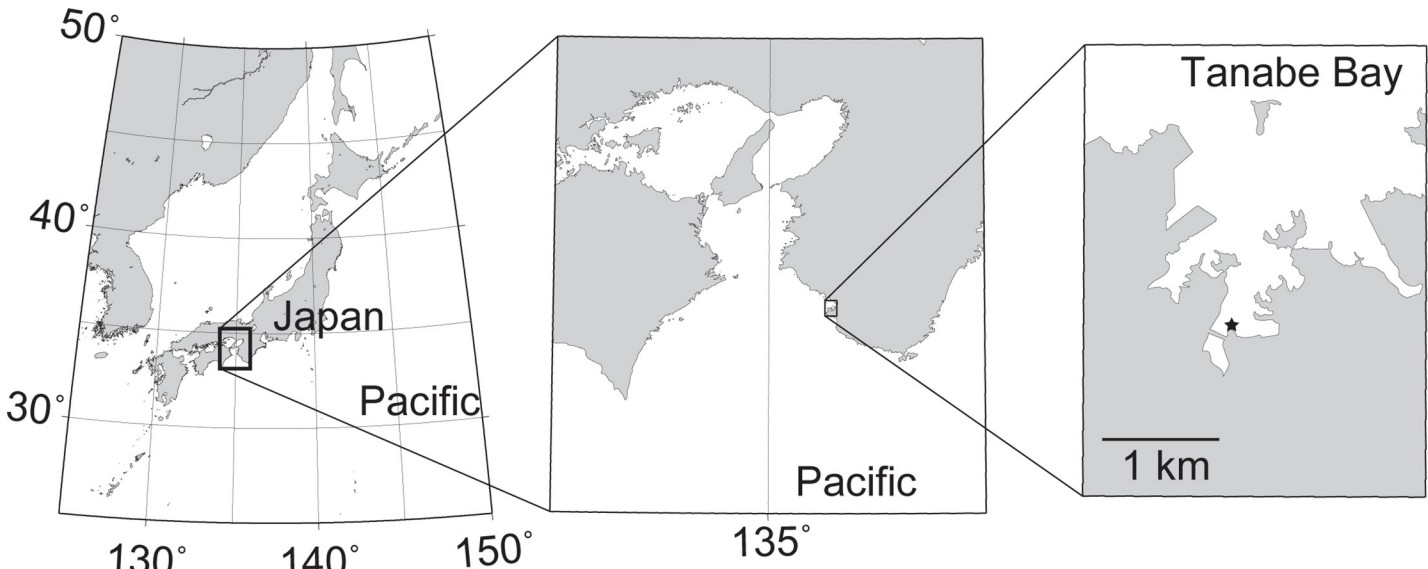

**Fig 1. Location of the sampling site in this study.** The black star in the right map shows the sampling site. These maps were drawn using the GMT [27].

half a century. Surface (1 m depth) and bottom seawaters (B−1 m depth; 1 m above bottom) were collected using a Van Dorn water sampler and pre-filtered through a 200-μm nylon mesh to remove mesozooplankton. Sediment core samples (7 cm in diameter) were taken in triplicate with a KK core sampler, being careful to avoid suspending the sediment surface. The soft boundary layer of each core sample (sediment depth = 0–1 mm) [26], which was easily resuspended with gentle shaking, was used as a resuspendable sediment sample for further analysis of the bacterial community structure. Each sample comprised approximately 50 mL of seawater.

### Environmental characteristics

Water temperature and salinity were measured using a modernized compact and lightweight multiparameter water quality meter, AAQ1183-H (JFE Advantec Co., Ltd., Hyogo, Japan). For chlorophyll $a$ (Chl $a$) measurements, we filtered 200 mL of water subsamples through GF/F filters (Whatman International Ltd., Maidstone, England) in duplicate. The Chl $a$ concentration was measured using a TD-700 fluorometer (Turner Designs, Sunnyvale, CA, USA) after extraction with $N,N$-dimethylformamide [28]. For particulate organic carbon (POC) measurements, 300–1,500 mL of water subsamples were filtered on precombusted (500°C, 3 h) GF/F filters in duplicate. For sediment organic carbon (SOC) measurements, surface layers of core samples in duplicate were used. POC and SOC were determined using a CHN analyzer, JM10 (J-SCIENCE LAB Co., Ltd., Kyoto, Japan).

To estimate the bacterial abundance in seawater, subsamples were fixed with 0.22 μm filtered paraformaldehyde (final concentration 2%), and filtered sequentially using 3.0 and 0.2 μm pore size Nuclepore™ membrane filters (Whatman International Ltd., Maidstone, England) under a < 100 mmHg vacuum. The filters were stained with 2 μg·mL$^{-1}$ 4',6-diamidino-2-phenylindole (DAPI) [29]. To estimate the sediment bacterial abundance, the surface layer subsample was fixed with 0.22 μm filtered paraformaldehyde (final concentration, 2%), and treated according to procedures described previously [30], with modifications. Briefly, sediment diluted with 0.01 M sodium pyrophosphate (Sigma-Aldrich Co. LLC, Saint Louis,

MO, USA) was vortexed for 45 min at 2,000 rpm using MixMate® (Eppendorf AG, Hamburg, Germany) and then sonicated for 30 s at 20 W (Ultrasonic Disruptors UD-211; Tomy Seiko Co. Ltd., Tokyo, Japan). Aliquots were filtered onto 0.2 μm pore Nuclepore™ membrane filters under a < 100 mmHg vacuum; subsequently, the filters were stained for 15 min with SYBR® Green I (1:500 dilution; Molecular Probes, Eugene, OR, USA). More than 300 cells per sample, or at least 10 microscopic fields, were counted using an epifluorescence microscope (BX51; Olympus Co., Tokyo, Japan) with U-MWU2 and U-MNIBA2 filters for DAPI and SYBR® Green I, respectively. We regarded bacteria on the 3.0 and 0.22 μm pore filters as particle-attached and free-living bacteria, respectively.

## BrdU labeling

Approximately 10 L of seawater samples and 50 mL of resuspendable sediment samples were incubated with BrdU (1 μM final concentration; Sigma-Aldrich Co. LLC) at ambient water temperatures (Table 1) for 5 h. After incubation, the bacterial cells in the seawater samples were collected with a Nuclepore™ membrane filter (3.0 μm pore size, 47 mm diameter; Whatman) and subsequently with a Sterivex™ cartridge filter (0.22 μm pore size; Millipore, MA, USA) using a peristaltic pump. The 3.0 μm Nuclepore™ membrane filters were changed 2–3 times before the filters became clogged. The bacterial cells in the sediment samples were collected with a Sterivex™ cartridge filter (0.22 μm pore size) using a syringe. Immediately after filtration, the filters were stored at −20˚C until further analysis.

## BUMP-DGGE analysis

This analysis was performed according to procedures previously described [24]. Briefly, the filters were subjected to xanthogenate-SDS DNA extraction and 0.5 or 1 μg of the extracted DNA was used for immunocapture. For the resuspendable samples, three samples were pooled after DNA extraction. Total and immunocaptured BrdU-labeled DNA samples were used as templates for PCR amplification of 16S rRNA genes using the eubacterial-specific primer 341F with a GC-clamp (5'-CGC CCG CCG CGC CCC GCG CCC GTC CCG CCG CCC CCG CCC GCC TAC GGG AGG CAG CAG-3', where the underlined letters indicate the GC-clamp) and the universal primer 907R (5'- CCG TCA ATT C[A/C]T TTG AGT TT-3') [31]. Approximately 200 ng of each PCR product was loaded onto a 6% polyacrylamide gel (acrylamide:$N$,$N$'-methylene bisacrylamide [37:1]; Nacalai Tesque, Kyoto, Japan) in 0.5 × TAE (20 mM Tris, 10 mM acetate, 0.5 mM Na$_2$EDTA, pH 8.2) with a denaturing gradient of 25% to 70% from top to bottom. Electrophoresis was performed at 85 V for 16 h at 60˚C in a hot-bath

**Table 1. Environmental characteristics at the sampling site.**

| Year | Month | Surface water[a] | | | | Bottom water | | | | Sediment |
|------|-------|------|------|------|------|------|------|------|------|------|
| | | WT | Sal | Chl $a$ | POC | WT | Sal | Chl $a$ | POC | SOC |
| | | (˚C) | (PSU) | (μg·L$^{-1}$) | (μg·L$^{-1}$) | (˚C) | (PSU) | (μg·L$^{-1}$) | (μg·L$^{-1}$) | (mg·g$^{-1}$ ww) |
| 2009 | May | 21.6 | 33.7 | 4.24 | 530.2 | 20.7 | 34.2 | 3.69 | 425.9 | 34.6 |
| | July | 26.1 | 33.3 | 1.71 | 485.6 | 24.7 | 33.6 | 0.69 | 271.9 | 25.7 |
| | September | 26.4 | 33.6 | 0.50 | 166.8 | 26.4 | 33.6 | 0.73 | 171.5 | 36.2 |
| | November | 19.0 | 33.3 | 0.72 | 175.5 | 20.0 | 33.3 | 0.45 | 264.5 | 25.2 |
| 2010 | January | 14.5 | 34.6 | 1.16 | 226.9 | 15.2 | 34.5 | 0.93 | 219.8 | 35.6 |
| | March | 14.0 | 32.5 | 2.95 | 320.3 | 15.1 | 33.5 | 5.21 | 398.5 | 21.9 |

[a] WT, water temperature; Sal, salinity; Chl $a$, chlorophyll $a$; POC, particulate organic carbon; SOC, sediment organic carbon; ww, wet weight

DGGE unit (Ingeny Inc., Goes, Netherlands) with a running buffer of $0.5 \times$ TAE. The gel was stained with $1 \times$ SYBR$^{\circledR}$ Gold (Molecular Probes, Eugene, OR, USA) in $0.5 \times$ TAE for 30 min in the dark, and then washed in $0.5 \times$ TAE. The gel was subsequently visualized and documented using ImageQuant 400 (GE Healthcare UK Ltd., Little Chalfont, UK).

Jaccard's coefficient was calculated based on the presence/absence of DGGE bands according to the formula: $S_{Jaccard} = N_{AB}/(N_A + N_B - N_{AB})$, where $N_{AB}$ is the number of common bands and $N_A$ and $N_B$ are the total number of bands in sample A and B, respectively [31]. The distance matrix was analyzed with the between-group average linkage method for clustering, using the 'Vegan' package [32] in R software (Vienna, Austria). We also performed a similarity profile routine (SIMPROF) with 9,999 permutations to test the significance ($p$ values) of the separated clusters [33] using the 'clustsig' package [34].

### Sequencing and phylogenetic analysis

Excised DGGE bands were sequenced directly from their PCR products, which were re-amplified with the primer set used above. Before sequencing, the PCR products were analyzed by DGGE to confirm the band positions relative to the original sample. Bidirectional sequencing using the 341F/907R primer set was performed by SolGent (South Korea, http://www.solgent.com/). The sequences were aligned to known sequences in the NCBI database using the standard nucleotide BLAST. Subsequent phylogenetic analysis was performed with MEGA 6 [35]. All sequences were validated with the Bellerophon program in Greengenes [36]. The nucleotide sequences were deposited into the DDBJ nucleotide sequence database under accession numbers LC314448 to LC314476.

## Results

### Environmental characteristics

The environmental characteristics of our sampling site are shown in Table 1. At both water depths, the water temperature exhibited a change with a maximum in September 2009 and minimum in March 2010, ranging from 14.0 to 26.4˚C. Salinity ranged from 32.5 to 34.6 PSU, without a clear pattern. The Chl $a$ concentration was relatively high in May and March and low in September and November, varying from 0.50 to 4.24 µg·L$^{-1}$ in the surface seawater and 0.45 to 5.21 µg·L$^{-1}$ in the bottom seawater. The POC concentration in the surface and bottom seawaters varied from 172.0 to 514.8 µg·L$^{-1}$ and 107.7 to 471.5 µg·L$^{-1}$, respectively. In sediment samples, the SOC concentration ranged from 21.9 to 36.2 mg·g$^{-1}$ wet weight. The abundance of free-living bacteria was higher than that of particle-attached bacteria at both water depths (Table 2). The cell concentration of free-living bacteria in the surface and bottom seawaters ranged from $6.8 \times 10^5$ to $2.8 \times 10^6$ cells·mL$^{-1}$ and $7.1 \times 10^5$ to $2.1 \times 10^6$ cells·mL$^{-1}$, respectively. In contrast, the cell concentrations of particle-attached bacteria in the surface and bottom seawaters ranged from $4.8 \times 10^3$ to $6.5 \times 10^4$ cells·mL$^{-1}$ and $4.2 \times 10^3$ to $6.2 \times 10^4$ cells·mL$^{-1}$, respectively. In sediment samples, the bacterial concentrations varied from $4.7 \times 10^7$ to $1.6 \times 10^8$ cells·g$^{-1}$ wet weight.

### Community structures of total and BrdU-incorporating bacteria

The DGGE profiles of PCR-amplified 16S rRNA genes from total DNA clearly show the bimonthly variation in the community structure of total bacteria during the studied period (Fig 2). A similar trend was observed in the profiles obtained from BrdU-incorporated DNA. The DGGE profiles separated into three main clusters (Fig 3) as follows: (1) May and November, (2) July and September, and (3) January and March clusters. Within the three main

**Table 2. Bacterial concentrations at the sampling site.**

| Year | Month | Surface water | | Bottom water | | Sediment |
|---|---|---|---|---|---|---|
| | | Free-living | Attached | Free-living | Attached | |
| | | $\times 10^5$ cells·mL$^{-1}$ | $\times 10^3$ cells·mL$^{-1}$ | $\times 10^5$ cells·mL$^{-1}$ | $\times 10^3$ cells·mL$^{-1}$ | $\times 10^7$ cells·g$^{-1}$ ww[b] |
| 2009 | May | 14.6 ± 2.5[a] | 64.6 ± 27.0 | 9.6 ± 1.4 | 62.0 ± 22.7 | 9.9 ± 2.5 |
| | July | 28.3 ± 4.6 | 6.3 ± 4.5 | 11.4 ± 3.0 | 7.0 ± 7.6 | 6.8 ± 1.4 |
| | September | 10.5 ± 1.2 | 27.2 ± 12.9 | 14.2 ± 1.8 | 24.9 ± 16.4 | 15.9 ± 3.7 |
| | November | 7.1 ± 1.0 | 36.3 ± 27.8 | 7.1 ± 1.2 | 27.5 ± 12.1 | 11.6 ± 2.6 |
| 2010 | January | 6.8 ± 1.0 | 4.8 ± 2.0 | 7.8 ± 0.9 | 4.2 ± 0.9 | 8.5 ± 2.2 |
| | March | 16.9 ± 2.2 | 6.4 ± 1.9 | 21.1 ± 3.0 | 5.3 ± 1.4 | 4.7 ± 1.3 |

[a] mean ± standard deviation

[b] ww, wet weight

clusters, the total and BrdU-incorporating communities formed different clusters (SIMPROF, $p < 0.05$) except for the samples obtained in May 2009. In May 2009, the BrdU-incorporating communities in the surface water cluster were classified in the same cluster as the total communities. The community structures of particle-attached bacteria were not distinguished from those of free-living bacteria in either the total or BrdU-incorporating communities (SIMPROF, $p > 0.05$). Although the total communities in the resuspendable sediments differed from those in the seawater (SIMPROF, $p < 0.05$), the BrdU-incorporating communities in the resuspendable sediments formed the same cluster as those in the seawater.

## Sequencing and phylogenetic analysis

Twenty-five of 27 were referred to as BrdU-incorporating phylotypes, including nine Alphaproteobacteria, five Gammaproteobacteria, one Deltaproteobacteria, seven Bacteroidetes, and three Actinobacteria (Fig 4, Table 3). All phylotypes related to Alphaproteobacteria belonged to the major subgroup Rhodobacterales, especially the *Roseobacter* clade. Throughout the studied period, Alphaproteobacteria was the most dominant bacterial group of the BrdU-incorporating communities, followed by Gammaproteobacteria and Bacteroidetes. Six phylotypes (four Rhodobacterales, Alphaproteobacteria; one Alteromonadales, Gammaproteobacteria; and one Actinomycetales, Actinobacteria) were detected at high frequencies (representing > 50% of the studied period) as BrdU-incorporating bacteria throughout the studied period. Phylotypes of Actinobacteria were the dominant group in September. Three phylotypes (two Rhodobacterales, Alphaproteobacteria; and one Flavobacteriales, Bacteroidetes) appeared only in BrdU-incorporating communities, whereas the two phylotypes related to Chloroflexi bacteria appeared only in total communities.

## Discussion

This is the first report to our knowledge determining the phylogenetic affiliations of AGB in resuspendable sediment and particle-attached samples, as well as free-living AGB, from an aquaculture environment. In this environment, large quantities of allochthonous organic matter, such as feed waste and fish feces, are present, with the degree of organic load depending on aquaculture activities [1, 5]. Thus, aquaculture activities drastically change the organic conditions [9]. Sinking particles and resuspendable sediments possess high rates of microbial hydrolysis and mineralization and are regarded as important sites for these activities [8, 9]. However, the types of bacteria responsible for these reactions have previously not been clearly elucidated. Delineating the phylogenetic affiliations of AGB that can adapt to such

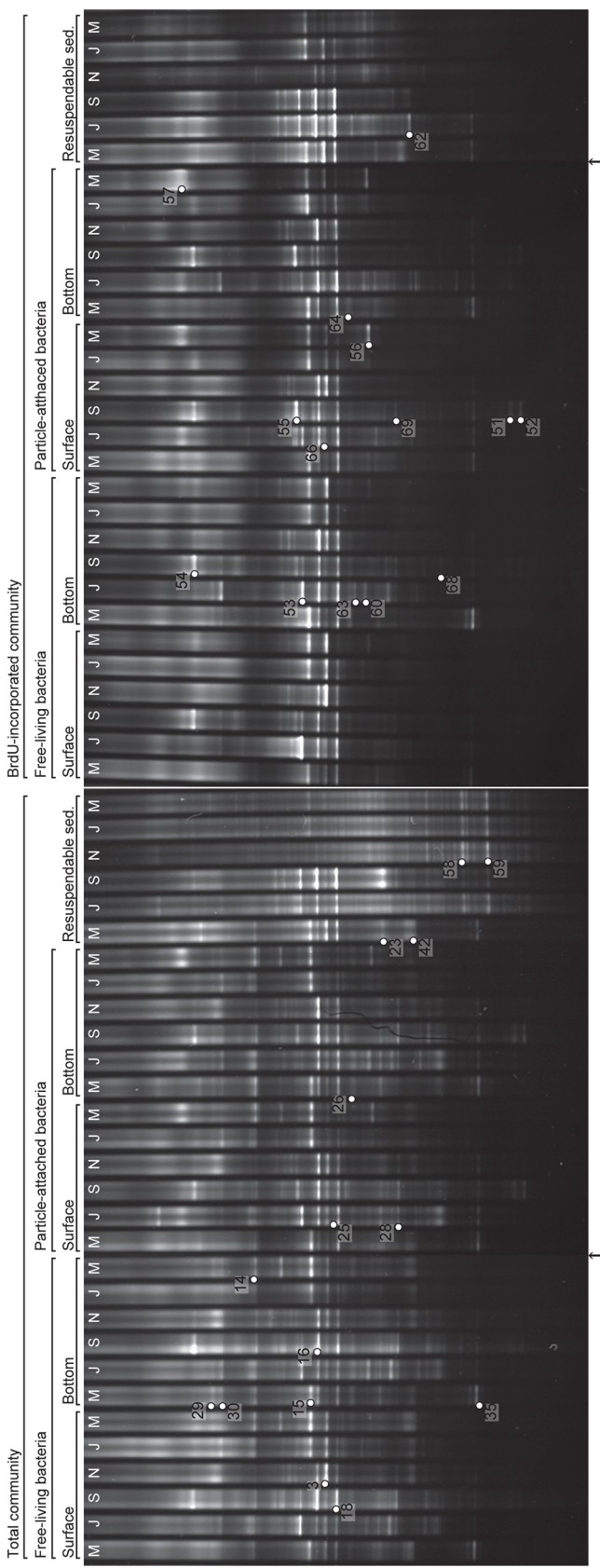

**Fig 2. Denaturing gradient gel electrophoresis images of 16S rRNA genes amplified from total and BrdU-incorporating communities in the aquaculture environment.** Arrows at the bottom of the two panels indicate where the image segments were joined because all the panels could not be imaged simultaneously. Numbers show the excised bands for sequencing analysis. Samples were collected in May (M), July (J), September (S), November (N), January (J), and March (M), and the corresponding lanes are labeled and ordered accordingly.

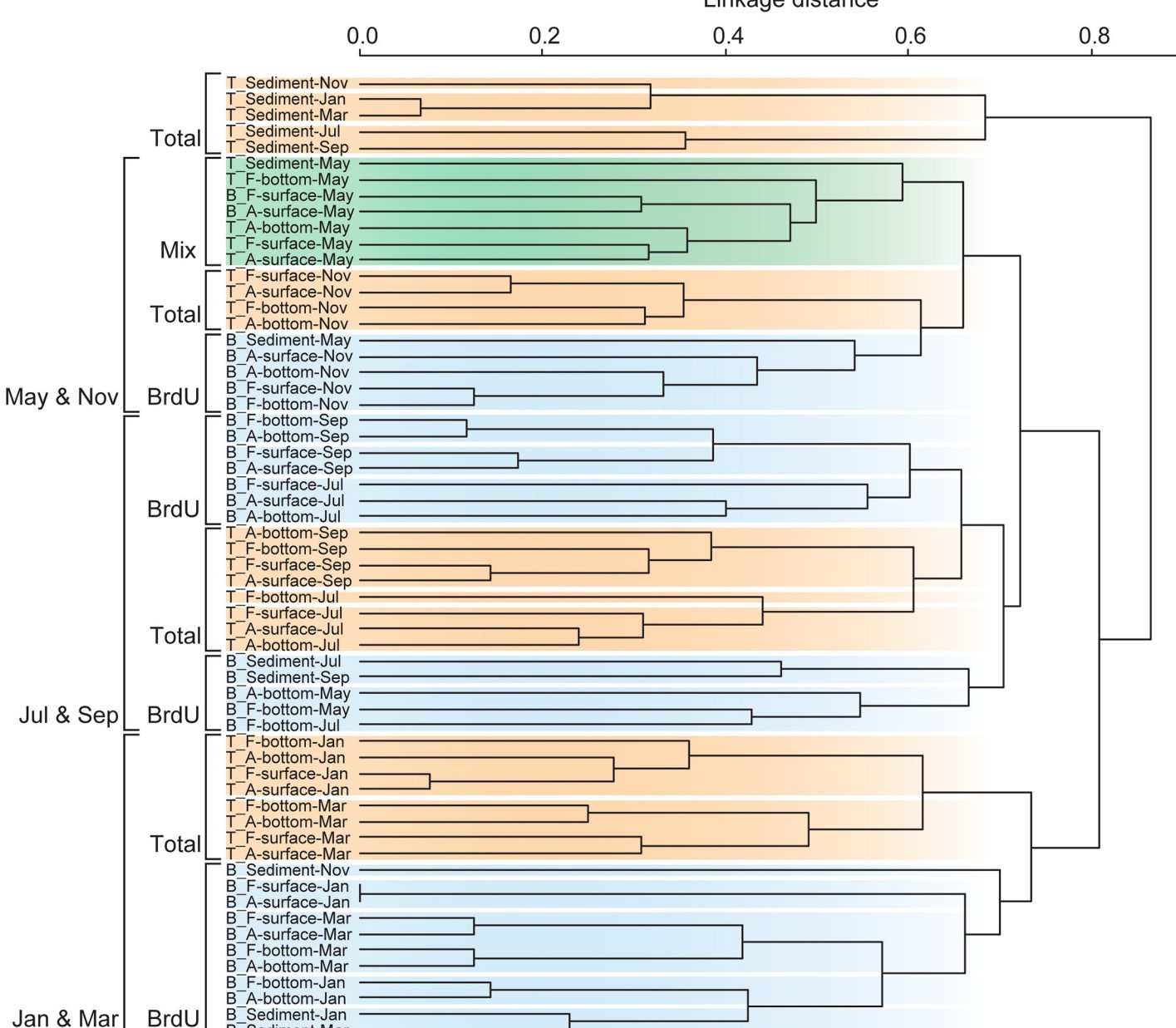

**Fig 3. Relationships among community structures of total and BrdU-incorporating bacteria from the aquaculture environment in the indicated months.** Each colored box (Orange: total community cluster, Blue: BrdU-incorporating community cluster, Green: total and BrdU-incorporating communities cluster) indicates distinguishable clusters (SIMPROF, $p < 0.05$). T: total community, B: BrdU-incorporating community, F: free-living bacterial community, A: particle-attached bacterial community.

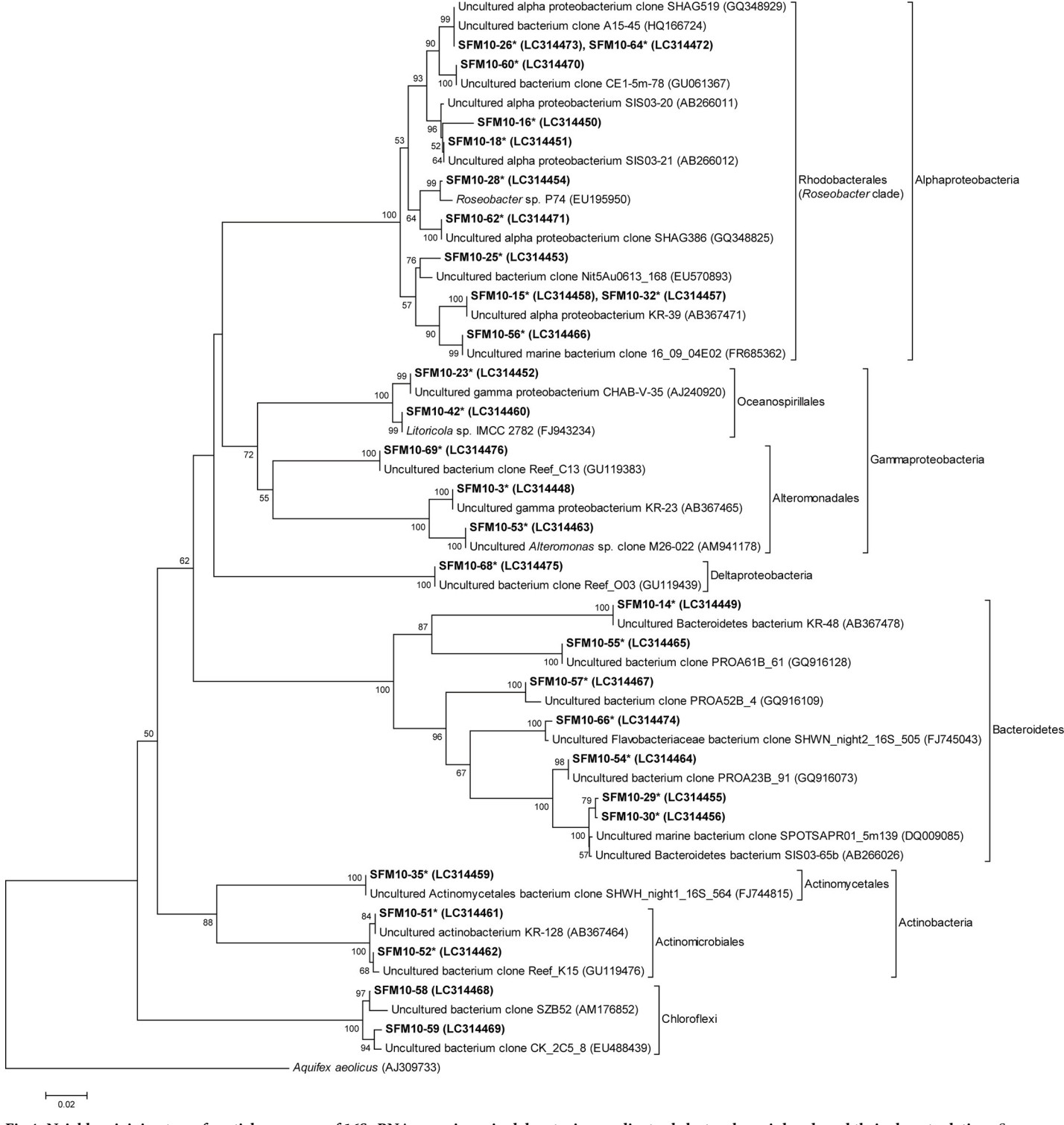

**Fig 4. Neighbor-joining tree of partial sequences of 16S rRNA genes in excised denaturing gradient gel electrophoresis bands and their closest relatives.** Sequences determined in this study are shown in bold. Bootstrap values (> 50%) are indicated by branches. The scale bar represents 2% estimated sequence divergence. Asterisks indicate the phylotypes found in BrdU-incorporating communities. *Aquifex aeolicus* was used as the outgroup.

**Table 3. The phylogeny of excised DGGE bands determined by 16S rRNA gene sequencing, and the presence or absence DGGE bands in samples.**

| SFM10 band no. | Accession no. | Phylogenetic group[a] | Free-living Surface water M | J | S | N | M | Free-living Bottom water M | J | S | N | M | Particle-attached Surface water M | J | S | N | M | Particle-attached Bottom water M | J | S | N | M | Resuspendable Sediment M | J | S | N | M |
|---|---|---|---|---|---|---|---|---|---|---|---|---|---|---|---|---|---|---|---|---|---|---|---|---|---|---|---|
| **Alphaproteobacteria** | | | | | | | | | | | | | | | | | | | | | | | | | | |
| 18 | LC314451 | Rhodobacterales | TB[b] | TB | TB | T | TB | TB | TB | TB | TB | TB | TB | TB | TB | TB | TB | TB | TB | TB | TB | B | B | TB | B | B | B |
| 16 | LC314450 | Rhodobacterales | TB | TB | TB | TB | - | TB | TB | TB | TB | - | TB | TB | TB | B | B | TB | TB | TB | B | B | TB | TB | TB | B | B |
| 26,64 | LC314473, LC314472 | Rhodobacterales | TB | TB | TB | T | TB | TB | TB | - | - | TB | TB | TB | - | - | T | TB | TB | - | T | - | - | - | - | T | TB |
| 15,32 | LC314458, LC314457 | Rhodobacterales | TB | - | - | - | TB | TB | - | - | TB | TB | TB | TB | - | TB | TB | TB | - | TB | TB | - | - | - | - | TB | TB |
| 56 | LC314466 | Rhodobacterales | - | - | - | B | - | B | - | - | - | B | B | B | - | B | TB | B | - | B | TB | - | - | - | - | TB | TB |
| 62 | LC314471 | Rhodobacterales | - | TB | B | - | - | - | TB | B | - | - | B | TB | - | - | - | TB | T | B | - | B | B | B | B | TB | - |
| 60 | LC314470 | Rhodobacterales | - | - | - | - | B | - | B | - | - | B | B | B | - | B | - | B | - | B | B | B | B | - | - | - | B |
| 25 | LC314453 | Rhodobacterales | - | - | - | - | - | - | - | - | B | B | B | B | B | B | - | B | B | B | - | B | B | B | B | - | B |
| 28 | LC314454 | Rhodobacterales | T | TB | T | T | T | T | TB | TB | T | - | T | TB | T | T | T | TB | TB | T | T | - | - | TB | - | - | B |
| **Gammaproteobacteria** | | | | | | | | | | | | | | | | | | | | | | | | | | |
| 3 | LC314448 | Alteromonadales | TB | TB | T | TB | TB | TB | T | T | TB | TB | TB | TB | T | T | B | TB | TB | TB | T | TB | B | TB | TB | B | TB |
| 53 | LC314463 | Alteromonadales | - | TB | TB | - | - | - | TB | B | - | - | - | B | B | B | B | B | B | B | B | - | TB | TB | B | - | - |
| 69 | LC314476 | Alteromonadales | T | TB | T | T | T | T | TB | TB | T | - | - | T | T | T | T | T | T | T | T | - | - | TB | - | - | B |
| 23 | LC314452 | Oceanospirillales | TB | T | T | T | TB | TB | TB | T | T | - | B | TB | T | T | - | B | TB | T | T | - | B | - | - | B | - |
| 42 | LC314460 | Oceanospirillales | T | - | - | T | T | T | - | TB | T | T | T | TB | T | T | T | T | TB | T | T | TB | TB | T | T | - | - |
| **Deltaproteobacteria** | | | | | | | | | | | | | | | | | | | | | | | | | | |
| 68 | LC314475 | uncultured | T | B | - | - | T | - | TB | - | - | B | - | B | - | B | B | B | B | B | B | - | - | B | - | TB | TB |
| **Bacteroidetes** | | | | | | | | | | | | | | | | | | | | | | | | | | |
| 14 | LC314449 | Flavobacteriales | TB | T | - | - | TB | TB | TB | B | - | TB | TB | TB | TB | TB | TB | TB | TB | T | TB | - | - | TB | - | - | - |
| 54 | LC314464 | Flavobacteriales | - | - | TB | T | - | - | B | TB | TB | - | - | - | - | B | B | - | - | B | B | - | - | - | B | B | - |
| 30 | LC314456 | Flavobacteriales | T | TB | TB | T | - | TB | TB | TB | TB | T | - | - | TB | TB | TB | T | TB | TB | T | - | T | T | T | - | - |
| 66 | LC314474 | Flavobacteriales | - | B | B | - | - | B | B | B | B | - | - | - | - | B | B | - | - | B | - | - | - | - | - | - | - |
| 29 | LC314455 | Flavobacteriales | T | - | - | - | TB | TB | - | - | - | - | T | T | T | TB | TB | T | - | TB | - | - | - | - | - | - | - |
| 57 | LC314467 | Cytophagales | - | - | B | - | - | - | TB | - | - | TB | B | TB | - | TB | - | TB | - | B | TB | - | - | - | B | B | - |
| 55 | LC314465 | uncultured | - | - | B | - | - | - | B | B | - | B | TB | - | TB | - | - | - | - | TB | - | B | B | - | B | B | - |
| **Actinobacteria** | | | | | | | | | | | | | | | | | | | | | | | | | | |
| 35 | LC314459 | Actinomycetales | TB | TB | TB | TB | - | TB | TB | TB | TB | TB | TB | - | TB | TB | TB | TB | TB | TB | TB | - | - | TB | TB | TB | - |
| 51 | LC314461 | Acidimicrobiales | - | B | TB | - | - | B | TB | TB | T | - | TB | - | TB | TB | TB | TB | TB | TB | - | - | - | - | - | B | - |
| 52 | LC314462 | Acidimicrobiales | - | - | TB | - | - | - | TB | TB | TB | - | TB | - | TB | TB | TB | - | TB | - | - | - | - | - | - | - | - |
| **Chloroflexi** | | | | | | | | | | | | | | | | | | | | | | | | | | |
| 58 | LC314468 | uncultured | - | - | - | - | - | - | - | - | - | - | - | - | - | - | - | - | - | - | T | T | T | T | - | T | T |
| 59 | LC314469 | uncultured | - | - | - | - | - | - | - | - | - | - | - | - | - | - | - | - | - | - | T | T | T | T | T | T | T |

[a] Order of the closest isolates. The band matched to no isolates with > 95% similarity is indicated as "uncultured" for convenience.

[b] T, presence of DGGE band in total communities; B, in BrdU-incorporating communities; TB, in both communities; -, DGGE band was not detected.

environments is important to understand the environmental capacity and carbon cycle in aquaculture environments.

Distinct community structures between total bacteria and AGB (Fig 3) were consistent with previous results from non-fish farming area [16, 24, 25]. This highlights the difficulty of determining which bacteria truly contribute to the organic matter cycle using conventional methods. The community structures of the free-living and particle-attached bacteria are similar, which may be owing to the POC and dissolved organic carbon concentrations [2, 8, 9]. In environments constantly affected by fish farming, the same bacterial species contribute to the degradation and utilization. Furthermore, the dominant bacterial species could be detected by DGGE, which analyzes the abundant species [31]. Further analysis focused on the identified bacteria in this study could effectively be used for sustainable fish farming because the dominant bacteria may contribute to organic matter degradation and cycling.

Previous studies have suggested that microbes in the sediment can act as a 'seed population' for those in the water column [37, 38]. However, our results do not agree with these postulates. The present study, analyzing a shallow fish-farming area, showed that bacteria originating from seawater were still actively growing when the sediment was resuspended (Fig 3). For example, Chloroflexi bacteria were found throughout the year in the resuspendable sediment of this aquaculture environment, but were not identified as AGB (Table 3). Although the sediment harbored a larger number of distinct bacteria, this was not considered to represent a "seed population" for AGB when the sediment was resuspended; instead, it was an alternative source of organic matter with a high organic carbon concentration. Thus, the carbon-rich organic matter of the resuspendable sediment could act as a supplement for bacterial growth in the water column.

All Alphaproteobacteria phylotypes associated with particles (Table 3) were closely related to the *Roseobacter* clade (Fig 4). This clade comprises dominant bacteria in coastal and open oceans, accounting for up to 20% of the total bacterial community [39]. *Roseobacter* includes aerobic anoxygenic phototrophs (AAnPs) that can use light as well as organic matter for energy and/or cellular biomass production, and accordingly have been classified as photoheterotrophs [40]. AAnPs appear to be abundant, especially in oligotrophic environments, owing to their photoheterotrophic property. However, previous studies have shown that they are not specifically adapted to oligotrophic environments, but are also abundant in eutrophic environments [41, 42]. Additionally, AAnPs are often found on particles [43]. Waidner and Kirchman [44] reported that most AAnPs are attached to particles (31% to 94% of total AAnPs). It was also reported that some *Roseobacter* bacteria in marine snow produce acylated homoserine lactones, suggesting that functions, such as biofilm formation and exoenzyme production on particles are regulated by a quorum sensing [45]. Therefore, further research is needed to reveal the ecological functions of particle-attached *Roseobacter* clade bacteria, which were identified as AGB in this study.

Bacteroidetes bacteria were also determined to be dominant particle-attached AGB (Table 3), in accordance with previous studies [10–13]. These bacteria are also dominant in both coastal and open oceans, accounting for more than half of the total bacterial cells in some reports, especially on particulate organic detritus [46]. Several studies have shown that Bacteroidetes are specialized in their attachment to particles [47, 48]. Through a comparative genomic approach, Fernández-Gómez et al. [49] revealed that Bacteroidetes bacteria have many advantages, such as adhesion and degradation of polymers and gliding motility, which allow them to grow on particles; in addition, they possess significantly more proteases than glycoside hydrolases. Considering these findings and the gene arrangement, it was concluded that members of Bacteroidetes are key players in particulate organic matter degradation, and especially protein-rich organic matter, owing to aquaculture activities. In addition, many *Flavobacteria*,

which are members of the Bacteroidetes phylum, contain proteorhodopsins, or light-driven proton pumps [50, 51]. These findings raise the possibility that the photoheterotrophic capacity of Bacteroidetes, as well as *Roseobacter* bacteria, are fundamental in the degradation of protein-rich particulate organic matter in aquaculture environments.

In the present study, Actinobacteria were identified as AGB (Table 3). Whereas Actinobacteria are generally considered as freshwater taxa [52], it has been reported that some species are also widely distributed in the ocean [53]. However, the ecological implication of marine Actinobacteria has not been fully clarified. Previous studies have shown that Actinobacteria phylotypes are actively growing in coastal and oceanic waters [16, 25], indicating that some of these species are well adapted to seawater. Known cultivable Actinobacteria in soil have the capacity to degrade diverse polymers, such as cellulose, lignin, chitin, and humic materials [54]. Although it is not clear if Actinobacteria identified as AGB in marine environments can degrade such polymers, these bacteria could contribute to the carbon cycle in our coastal fish-farming area as well as the ocean.

Eight phylotypes (SFM10-16, SFM10-18, SFM10-15 and -32, and SFM10-26 and -64 of Alphaproteobacteria; SFM10-3 of Gammaproteobacteria; SFM10-14, SFM10-29, and SFM10-30 of Bacteroidetes) were closely related to active growers in other coastal seawaters [24, 25, 55]. Among them, five phylotypes (SFM10-16, SFM10-18, SFM10-15 and -32, SFM10-26 and -64, and SFM10-3) were present at a higher frequency (> 50%) in the BrdU-incorporating community (Table 3). These phylotypes might contribute to the carbon cycle both in this aquaculture and ordinary coastal environments through their active organic matter degradation and/or utilization. Different AGB species utilize diverse amounts and types of organic matter, which can change organic matter fluxes. These phylotypes should be targeted as key coastal species, and their dynamics and ecological functions should be monitored.

In the present study, we reported novel information regarding AGB in a coastal shallow fish-farming area. Although total community structures were considerably different between the water column and resuspendable sediment, the active growers were similar. This result suggests that bacterial species contributing to the environmental capacity and carbon cycle in the water column are limited. Focusing on the limited bacterial species, we can more accurately and precisely assess the environmental capacity and carbon cycle, not only in aquaculture areas, but also in coastal environments.

## Supporting information

**S1 Table. Raw binary data (presence/absence of DGGE bands) used for cluster analysis.**
(XLSX)

**S1 Raw images.**
(PDF)

## Acknowledgments

We would like to thank Osamu Murata and Keitaro Kato, Aquaculture Research Institute, Kindai University, for their help, as well as Sosuke Hirata for his support in sampling and laboratory analyses. We would like to thank Editage (www.editage.com) for English language editing.

## Author Contributions

**Conceptualization:** Akito Taniguchi, Mitsuru Eguchi.

**Data curation:** Akito Taniguchi.

**Formal analysis:** Akito Taniguchi.

**Funding acquisition:** Mitsuru Eguchi.

**Investigation:** Akito Taniguchi.

**Methodology:** Akito Taniguchi.

**Project administration:** Mitsuru Eguchi.

**Resources:** Akito Taniguchi.

**Supervision:** Mitsuru Eguchi.

**Visualization:** Akito Taniguchi.

**Writing – original draft:** Akito Taniguchi.

**Writing – review & editing:** Mitsuru Eguchi.

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
