## [Decision Letter · Decision Letter 0]

9 Mar 2020

PONE-D-19-35595

Community structure of actively growing bacteria in a coastal fish-farming area

PLOS ONE

Dear Dr. Taniguchi,

Thank you for submitting your manuscript to PLOS ONE. After careful consideration, we feel that it has merit but does not fully meet PLOS ONE’s publication criteria as it currently stands. Therefore, we invite you to submit a revised version of the manuscript that addresses the points raised during the review process.

While most points must be addressed, particular attention should be paid to the lack of any information on the statistical methods in the methods section. For example there is no analysis of any temporal patterns and no analysis of environmental-bacteria relationships.  Although cluster analysis is obvious in the text later on, it is not described in the methods and we have no idea of either the cluster method nor the distance measure used.

We would appreciate receiving your revised manuscript by Apr 23 2020 11:59PM. To enhance the reproducibility of your results, we recommend that if applicable you deposit your laboratory protocols in protocols.io, where a protocol can be assigned its own identifier (DOI) such that it can be cited independently in the future. For instructions see: http://journals.plos.org/plosone/s/submission-guidelines#loc-laboratory-protocols

We look forward to receiving your revised manuscript.

Kind regards,

Judi Hewitt

Academic Editor

PLOS ONE

Journal Requirements:

Reviewers' comments:

Reviewer's Responses to Questions

**Comments to the Author**

1. Is the manuscript technically sound, and do the data support the conclusions?

Reviewer #1: Partly

Reviewer #2: Yes

2. Has the statistical analysis been performed appropriately and rigorously? 

Reviewer #1: No

Reviewer #2: Yes

3. Have the authors made all data underlying the findings in their manuscript fully available?

Reviewer #1: Yes

Reviewer #2: Yes

4. Is the manuscript presented in an intelligible fashion and written in standard English?

Reviewer #1: No

Reviewer #2: Yes

5. Review Comments to the Author

Reviewer #1: Introduction

Line 46 – Would be beneficial for the reader to define what do you mean by environmental capacity in this context?

Line 55 – rephrase: Previous studies looking at the community structures of particle-attached bacteria [9-12] showed that Gammaproteobacteria and the Bacteroidetes frequently dominated these particles.

Line 56 – rephrase: Further, most studies have focused on describing the general phylogenetic affiliations of these particle-attached bacteria, but information on the actively growing species remain little.

Line 62 – remove word ‘should’

Line 64 – replace protozoa with ‘heterotrophic or mixotrophic protists.’ Protozoa is not used anymore in microbial ecology

Line 71 – rephrase: For example, the diversity and spatio-temporal variability of AGB were investigated in coastal and oceanic environments using BrdU magnetic bead immunocapture coupled with PCR-denaturing gradient gel electrophoresis (BUMP-DGGE) techniques.

Line 76 – the shift to the next paragraph seems to be very abrupt. What’s the knowledge gap in terms of the technique use in aquaculture?

Line 78 – rephrase: The objectives of the present study then were to utilize this BrdU technique to determine the phylogenetic affiliations of AGB in the subsurface (ca. 10 m) of a coastal shallow fish-farming area to reveal their seasonal variations. We specifically looked at the free-living and particle-attached AGB in the water column.

Line 80-83 – the sentence seems to not fit.

The ending of this paragraph seems abrupt and not well structured. So, what are the new knowledge and insights? Why is this study actually important?

Materials and Methods

Line 93 – what is the goal why do bimonthly sampling? Justify.

Line 97 – ‘using a van dorn water sampler’

Line 124-135 – move this justification before describing the serial filtration

Line 140 – what were the in-situ water temperatures?

Line 175- Solgent based in what country?

Line 177 – what particular BLAST method?

Results

Line 187 – what’s a regular pattern? Or you meant seasonal patterns?

When you describe the patterns, what were the seasons identified? What characteristics do you associate with what season? What were the markers of these seasons? Be more specific in describing the results.

Table 1 – did you only measure once for each sample? Don’t the values should have standard errors?

Line 218 – so what the ‘variation’ that the communities clearly showed?

Figure 2 – Wouldn’t the data better presented in an ordinate system?

Line 227 – resuspendable or resuspended?

Line 245-248 – remove the 1st and 2nd sentences as they are not needed, just a repeat of the methods.

Line 248 – you mentioned of 29 bands, but why only 25 out of the 27? What are the other 2?

Line 257 – remove ‘bacteria throughout the studied period’. Redundant.

Table 3 can be moved as a supplementary material.

Discussions

Line 279 - In this environment, large quantities of allochthonous organic matter such as from feed wastes and fish feces are present, with the degree of organic load depending on aquaculture activities.

Line 285 – again, what do you mean by ‘environmental capacity’? or are you referring to ecosystem functions?

Line 290 – ‘were still actively growing when the sediments….’

Line 307-309 – why are they usually attached to particles?

Major comment; Roseobacter is one of the most studied bacterial groups, with many genome representatives. The recommendation to study them is too broad. What were the characteristics that allowed them to be attached? Why are they usually attached? Any suggestions from the literature?

Line 313- ‘also determined’

Line 315 – contextualize in what oceans and conditions?

Line 320 – so what if they have more proteases then glycoside hydrolases? What’s the implication of these to the particle?

Line 343 – how do they contribute to the ‘environmental capacity’ and carbon cycling? These are very general descriptions with no substantiation.

Other major comments

- There were no discussions why there was separate clustering between the total and binding, sediment vs. pelagic.

- The patterns of seasonality were not discussed especially relative to potential community drivers.

- There were no insights on the succession of the actively growing bacteria spatially and temporally, which were included in the design of the study.

- The discussions were more like a review of literature with limited insights and interpretations of generated results.

- Limited analysis especially on ecological statistics.

- Overall, the objectives were not answered or discussed accordingly particularly the 1) seasonality, and 2) the resuspension via waves and currents since there were no data generated for this.

- the manuscript will benefit from edits and improvement of the language used.

Reviewer #2: This study provides some information about the bacterial population present in the water column and sediment of a fish-farm area including both free-living and particle attached and the actively growing part of the population.the study is rigorous and well presented and the results are based on correct data. My only concern is that the objective and the application and interest of the study is not well described. I suggest the authors to explain the specific interest of the study and its contribution to the analysis, managing and monitoring and of the aquaculture areas

6. PLOS authors have the option to publish the peer review history of their article (what does this mean?). If published, this will include your full peer review and any attached files.

Reviewer #1: No

Reviewer #2: No

---

## [Author Response · Author response to Decision Letter 0]

23 May 2020

While most points must be addressed, particular attention should be paid to the lack of any information on the statistical methods in the methods section. For example there is no analysis of any temporal patterns and no analysis of environmental-bacteria relationships. Although cluster analysis is obvious in the text later on, it is not described in the methods and we have no idea of either the cluster method nor the distance measure used.

-Thank you for the suggestion. In response to the suggestion, a revised paragraph has been added to the Materials and Methods (lines 166–172).

Review Comments to the Author

-Thank you for the valuable comments and suggestions.

Reviewer #1: Introduction

Line 46 – Would be beneficial for the reader to define what do you mean by environmental capacity in this context?

-We have added the definition (lines 43–48).

Line 55 – rephrase: Previous studies looking at the community structures of particle-attached bacteria [9-12] showed that Gammaproteobacteria and the Bacteroidetes frequently dominated these particles.

-Thank you for the advice. We have rephrased this and the revised text has been proofread by a professional editing service (Editage) (lines 57–59).

Line 56 – rephrase: Further, most studies have focused on describing the general phylogenetic affiliations of these particle-attached bacteria, but information on the actively growing species remain little.

-We have rephrased this and the revised text has been proofread by a professional editing service (Editage) (lines 59–61).

Line 62 – remove word ‘should’

-We have removed the word (line 64).

Line 64 – replace protozoa with ‘heterotrophic or mixotrophic protists.’ Protozoa is not used anymore in microbial ecology

-Thank you for the advice. We have replaced it as suggested (line 66).

Line 71 – rephrase: For example, the diversity and spatio-temporal variability of AGB were investigated in coastal and oceanic environments using BrdU magnetic bead immunocapture coupled with PCR-denaturing gradient gel electrophoresis (BUMP-DGGE) techniques.

-We have rephrased this and then have it proofread by a professional editing service (Editage) (lines 73–76).

Line 76 – the shift to the next paragraph seems to be very abrupt. What’s the knowledge gap in terms of the technique use in aquaculture?

-Thank you for the suggestion. In accordance with the suggestion, we have added the following sentence at the end of this paragraph (lines 77–79): “Using this technique, it is possible to identify key bacterial species that directly contribute to degradation and/or mineralization of organic matter and to ensure sustainable aquaculture environments.”

Line 78 – rephrase: The objectives of the present study then were to utilize this BrdU technique to determine the phylogenetic affiliations of AGB in the subsurface (ca. 10 m) of a coastal shallow fish-farming area to reveal their seasonal variations. We specifically looked at the free-living and particle-attached AGB in the water column.

-We have rephrased this and the revised text has been proofread by a professional editing service (Editage) (lines 80–82).

Line 80-83 – the sentence seems to not fit.

-We have deleted the sentence.

The ending of this paragraph seems abrupt and not well structured. So, what are the new knowledge and insights? Why is this study actually important?

-Thank you for the suggestion. We have added the following sentence at the end of this paragraph (lines 86–88): “This is the first study that the AGB, not inactive and/or dead bacterial cells, which can directly contribute to the organic matter cycle of fish-farming areas, have been identified.”

Materials and Methods

Line 93 – what is the goal why do bimonthly sampling? Justify.

-In the study area, the temperature of the surface seawater ranges from 12–15°C (January–March) to 26–28°C (July–September). Thus, we sampled in the coldest, warmest and intermediate months. 

Line 97 – ‘using a van dorn water sampler’

-We have retained this name because the Van Dorn water sampler is named after developer (line 97) and it is common practice to use the full name of the water sampler. 

Line 124-135 – move this justification before describing the serial filtration

-We have not moved these sentences because they refer to sediment samples, not seawater samples (lines 123–132). 

Line 140 – what were the in-situ water temperatures?

-The “in situ water temperatures” refers to the water temperatures measured at the time of sampling. We have changed the word “in situ” to “ambient” and have referred to Table 1 (line 140). 

Line 175- Solgent based in what country?

-We have added the country, South Korea (line 178).

Line 177 – what particular BLAST method?

-We used the standard nucleotide BLAST in NCBI (line 180). 

Results

Line 187 – what’s a regular pattern? Or you meant seasonal patterns?

When you describe the patterns, what were the seasons identified? What characteristics do you associate with what season? What were the markers of these seasons? Be more specific in describing the results.

-Thank you for the suggestion. We have revised this (lines 188–193). We meant a regular pattern, such as the pattern of water temperature, which we used (along with air temperature) to identify seasons. We have toned down any focus seasonality because what we wanted to reveal in this manuscript was which bacterial species were active growers in the fish-farming area and whether the AGB was different for free-living, particle and resuspendable sediment.

Table 1 – did you only measure once for each sample? Don’t the values should have standard errors?

-We measured this according to traditionally used methods (e.g. see the section “Chlorophyll a levels, phytoplankton cell counts, and POC” in the Material and Methods in Riemann, Steward & Azam, 2000, Applied and Environmental Microbiology, 66:578–587). We measured water temperature and salinity using a highly accurate device, while Chl a, POC and SOC were measured in duplicate as described in the Materials and Methods. The values given are the averages.

Line 218 – so what the ‘variation’ that the communities clearly showed?

-We have changed “variation” to “bimonthly variation” (line 221).

Figure 2 – Wouldn’t the data better presented in an ordinate system?

-We have revised this (Fig. 2).

Line 227 – resuspendable or resuspended?

-We use the word “resuspendable” in this study since we resuspended the sediment samples with gentle shaking.

Line 245-248 – remove the 1st and 2nd sentences as they are not needed, just a repeat of the methods.

-We have removed these sentences. 

Line 248 – you mentioned of 29 bands, but why only 25 out of the 27? What are the other 2?

-This is because band no. 26 and 15 belonged to the same phylotypes of band no. 64 and 32, respectively (Table 3). 

Line 257 – remove ‘bacteria throughout the studied period’. Redundant.

-We have removed these words.

Table 3 can be moved as a supplementary material.

-We have kept Table 3 in the manuscript because in it we describe the presence and absence of the phylotypes and their frequency, which are important results. 

Discussions

Line 279 - In this environment, large quantities of allochthonous organic matter such as from feed wastes and fish feces are present, with the degree of organic load depending on aquaculture activities.

-We have rephrased this and the revised text has been proofread by a professional editing service (Editage) (lines 281–283).

Line 285 – again, what do you mean by ‘environmental capacity’? or are you referring to ecosystem functions?

-We have added a definition for ‘environmental capacity’ (lines 45–48).

Line 290 – ‘were still actively growing when the sediments….’

-We have added the word “still” (line 303).

Line 307-309 – why are they usually attached to particles?

-There are two possibilities. The first is that AAnPs have extracellular matrices that may facilitate attachment to particles, and the second is that the reduced oxygen concentration in particles where AAnPs would be unaffected is due to their phototrophy (Waindner&Kirchman, Appl Environ Microbiol, 2007; 73:3936–3944). 

Major comment; Roseobacter is one of the most studied bacterial groups, with many genome representatives. The recommendation to study them is too broad. What were the characteristics that allowed them to be attached? Why are they usually attached? Any suggestions from the literature?

-Thank you for the meaningful comment. We think that this may be due to our inability to analyze actively growing Roseobacter alone with the intention of attaching to particles by most conventional techniques. What we mean is that it is hard to find which Roseobacter should be focused on because there are many unrelated Roseobacter identified by such techniques. By identifying the specific Roseobacter and the attachment mechanisms, in addition to using laboratory experiments, we can see the reason why. We do think that one useful approach is the BrdU technique, which can identify the phylotypes of the Roseobacter group, but it may be overestimated in the data from this study. Thank you very much for your meaningful comment.

Line 313- ‘also determined’

-We have revised this (line 326).

Line 315 – contextualize in what oceans and conditions?

-We have revised the sentence (lines 327–329).

Line 320 – so what if they have more proteases then glycoside hydrolases? What’s the implication of these to the particle?

-We have revised the sentence (lines 334–336). In aquaculture environments, there is a lot of protein-rich organic matter derived from feed, so the role of Bacteroidetes should be more important.

Line 343 – how do they contribute to the ‘environmental capacity’ and carbon cycling? These are very general descriptions with no substantiation.

-We have revised the sentence (lines 357–359). They might contribute to carbon cycling because they require increased levels of organic carbon to maintain their cell production and are easily lysed by viruses and/or grazed by heterotrophic or mixotrophic protists. Thus, they should significantly contribute to the environmental capacity by actively degrading organic matter.

Other major comments

- There were no discussions why there was separate clustering between the total and binding, sediment vs. pelagic.

-Thank you for the suggestion. We have added the following paragraph (lines 290–299).

“Distinct community structures between total bacteria and AGB (Fig. 3) were consistent with previous results from non-fish farming area [16, 24, 25]. This highlights the difficulty of determining which bacteria truly contribute to the organic matter cycle using conventional methods. The community structures of the free-living and particle-attached bacteria are similar, which may be owing to the POC and dissolved organic carbon concentrations [2, 8, 9]. In environments constantly affected by fish farming, the same bacterial species contribute to the degradation and utilization. Furthermore, the dominant bacterial species could be detected by DGGE, which analyzes the abundant species [31]. Further analysis focused on the identified bacteria in this study could effectively be used for sustainable fish farming because the dominant bacteria may contribute to organic matter degradation and cycling.”

- The patterns of seasonality were not discussed especially relative to potential community drivers.

-Thank you for the suggestion. As you have pointed out several times, the seasonal variability of AGB may be overestimated. What we wanted to reveal in the manuscript was which bacterial species were active growers in this fish-farming area and whether the AGB was different for free-living, particle and resuspendable sediment. We have toned down our statement about seasonality throughout the manuscript. 

- There were no insights on the succession of the actively growing bacteria spatially and temporally, which were included in the design of the study.

-We have toned down our statement about seasonal (temporal) variability in the revised manuscript. On the other hand, we investigated spatial variability in the manuscript (that is, free-living (water), particle and resuspendable sediment). We have cited previous studies using the BrdU techniques in the Introduction of the study.

- The discussions were more like a review of literature with limited insights and interpretations of generated results.

-In the present study, we wanted to reveal the differences of AGB among free-living, particle-attached and resuspendable sediments. Therefore, the Discussion focused on what bacteria phylotypes were actively growing and where they were found in previous studies. Based on the findings obtained in this study, we are planning detailed experiments such as microcosm experiments. We think that the detailed function and role of each phylotype identified can be understood. 

- Limited analysis especially on ecological statistics.

-What we wanted to reveal in this manuscript was whether the AGB was different for free-living, particle and resuspendable sediment. In the revised manuscript, we have toned down the focus seasonality, as you suggested. Therefore, we think that the cluster analysis and the significance analysis for the clusters have been matched.

- Overall, the objectives were not answered or discussed accordingly particularly the 1) seasonality, and 2) the resuspension via waves and currents since there were no data generated for this.

-Thank you for the comment. As explained above, we have toned down our statement about seasonality throughout the revised manuscript. We have no data on resuspension via waves and currents in this farming-area, as you mention, but we were able to collect the bottom-seawater sample with sediment particles several times during sampling events. The possibility of resuspension might be so high that it is meaningful to show the potential for bacterial growth in this study. In future research, we would like to investigate how much resuspension actually occurs.

- the manuscript will benefit from edits and improvement of the language used.

-This manuscript had been already edited by a professional editing service (Editage; https://www.editage.jp/) before the submission (we have attached the certificate of English editing). The revised manuscript was edited again by Editage before its resubmission.

Reviewer #2: This study provides some information about the bacterial population present in the water column and sediment of a fish-farm area including both free-living and particle attached and the actively growing part of the population.the study is rigorous and well presented and the results are based on correct data. My only concern is that the objective and the application and interest of the study is not well described. I suggest the authors to explain the specific interest of the study and its contribution to the analysis, managing and monitoring and of the aquaculture areas

-Thank you for your comments. We have modified the explanation of the objective where appropriate in the revised manuscript (lines 57–61, 80–88). The specific interest of our study was to determine which bacterial species were active growers in this fish-farming area and whether the AGB was different for free-living, particle and resuspendable sediment. The fine-scale dynamics of the key bacterial species identified by our BrdU techniques will be investigated with both time-series sampling and mesocosm experiments in the future, and the findings are expected to help keep aquaculture environmentally sustainable.

---

## [Editor Report · Decision Letter 1]

15 Jun 2020

Community structure of actively growing bacteria in a coastal fish-farming area

PONE-D-19-35595R1

Dear Dr. Taniguchi,

We’re pleased to inform you that your manuscript has been judged scientifically suitable for publication and will be formally accepted for publication once it meets all outstanding technical requirements.

Kind regards,

Judi Hewitt

Academic Editor

PLOS ONE
---

## [Editor Report · Acceptance letter]

18 Jun 2020

PONE-D-19-35595R1 

Community structure of actively growing bacteria in a coastal fish-farming area 

Dear Dr. Taniguchi:

I'm pleased to inform you that your manuscript has been deemed suitable for publication in PLOS ONE. Congratulations! Your manuscript is now with our production department. 

Kind regards, 

on behalf of

Dr. Judi Hewitt 

Academic Editor

PLOS ONE